# The Impact of a Precision-Based Exercise Intervention in Childhood Hematological Malignancies Evaluated by an Adapted Yo-Yo Intermittent Recovery Test

**DOI:** 10.3390/cancers14051187

**Published:** 2022-02-25

**Authors:** William Zardo, Emanuele Villa, Eleonora Corti, Tommaso Moriggi, Giorgia Radaelli, Alessandra Ferri, Mauro Marzorati, Cristiano Eirale, Paola Vago, Andrea Biondi, Momcilo Jankovic, Adriana Balduzzi, Francesca Lanfranconi

**Affiliations:** 1Maria Letizia Verga Center, Department of Pediatrics, Università degli Studi di Milano-Bicocca, ASST Monza/MBBM Foundation, 20900 Monza, Italy; lelevilla94@hotmail.it (E.V.); cortieleonora1@gmail.com (E.C.); tommasomoriggi@gmail.com (T.M.); radaelli.personal@gmail.com (G.R.); andrea.biondi@unimib.it (A.B.); momcilo@libero.it (M.J.); abalduzzi@fondazionembbm.it (A.B.); francesca.lanfranconi@unimib.it (F.L.); 2Institute for Health and Sport, Victoria University, Melbourne 3011, Australia; alessandra.ferri@vu.edu.au; 3Institute of Biomedical Technologies, National Research Council, 20054 Segrate, Italy; mauro.marzorati@itb.cnr.it; 4Paris Saint Germain Football Club, 78100 Paris, France; ceirale@psg.fr; 5Department of Pedagogy, Università Cattolica del Sacro Cuore, 20123 Milano, Italy; paola.vago@unicatt.it

**Keywords:** hematology, pediatric, performance, Yo-Yo test, precision-based exercise, leukemia, lymphoma, hematopoietic stem cell transplantation

## Abstract

**Simple Summary:**

Disability is a temporary phenomenon for every child, adolescent, and young adult with hematological malignancies during the intensive phases of cancer treatment, but it can become a long-lasting condition for many. Disability is an umbrella term for impairments, activity limitations and participation restrictions, denoting the negative aspects of the interaction between an individual and that individual’s contextual, environmental, and personal factors. Adapted precision-based training programs during cancer treatment are an emerging therapeutic option in pediatric oncology and evaluating the impact of tailored exercise on individuals’ performance is mandatory for adapting exercise/sports activities. Our research showed that a new intermittent and recovery test, the Yo-Yo AD, provided valid information on an individual’s capacity to perform repeated intense exercise and to follow up on the impact of precision-based exercise intervention in childhood hematological malignancies.

**Abstract:**

During cancer treatments in childhood hematological malignancies, reduced exercise tolerance is one of the main hardships. Precision-based training programs help children, adolescents, and young adults and their families to resume regular physical activity, exercise, and sports once they return to their communities after the intensive phases spent in hospital. This study was aimed at verifying whether an intermittent recovery test, the Yo-Yo AD, could provide a simple and valid way to evaluate an individual’s capacity to perform repeated intense exercise and to follow up on the impact of tailored exercise in children, adolescents, and young adults with hematological malignancies. The Yo-Yo AD involved the repetition of several shuttles to muscle exhaustion, at pre-established speeds (walking and slow running). The heart rate (HR) and oxygen saturation (SaO_2_) were monitored during the test. The total distance and the walking/running ability, measured as the slope of the HR vs. distance correlation, were investigated before (T0) and after 11 weeks (T1) of precision exercise intervention. The Yo-Yo AD was also performed by healthy children (CTRL). Ninety-seven patients (10.58 ± 4.5 years, 46% female) were enrolled. The Yo-Yo AD showed the positive impact of the exercise intervention by increasing the distance covered by the individuals (T0 = 946.6 ± 438.2 vs. T1 = 1352.3 ± 600.6 m, *p* < 0.001) with a more efficient walking/running ability (T0 = 2.17 ± 0.84 vs. T1 = 1.73 ± 0.89 slope, *p* < 0.0164). CTRLs performed better (1754.0 ± 444.0 m, *p* = 0.010). They were equally skillful (1.71 ± 0.27 slope) when compared to the patients after they received the precision-based intervention. No adverse events occurred during the Yo-Yo AD and it proved to be an accurate way of correctly depicting the changes in performance in childhood hematological malignancies.

## 1. Introduction

During cancer treatment, children, adolescents, and young adults with hematological malignancies (CAYA-H) face reduced exercise tolerance, which adds a further burden to their health [1,2,3]. Furthermore, CAYA-H’s fate is to become accustomed to decreasing social inclusion opportunities, notably those related to participation in sports training and competition, due to their impaired individual exercise capacity and immunological status [3]. Exercise capacity is still reduced at the end of cancer treatment, after individuals are off therapy for 1 to 5 years, and even after 15 years of follow-up, ultimately limiting individuals’ full self-realization in society and jeopardizing their transition towards adulthood [4,5,6,7,8]. CAYA-H and their families all over the world share a common path of uncertainty after surviving cancer, and experience compromised exercise ability in the long term. As a consequence, their prospects of finding suitable workplaces and/or of performing physical activities, exercises, training, and sports are dramatically narrowed [7].

In 2030, the expected number of CAYA-H in Europe will be around 750,000, but the remarkable resilience of CAYA-H must nonetheless be tested by the legacy of cancer treatment, i.e., chronic diseases impacting their exercise tolerance and quality of life [9,10]. CAYA-H have a higher risk of developing cardiovascular disease [11], reduced pulmonary function [12,13,14,15,16], impaired skeletal muscle oxidative capacity [4,17], and impaired motor nervous function [18]. Furthermore, their reduced bone mineral density and/or blood supply increases their risk of developing osteoporosis [19,20] and osteonecrosis [21,22,23,24]. Each CAYA-H will experience disability directly at some point in his or her life during the intensive phases of cancer treatment; this disability can become a long-lasting condition for many [1,2].

As stated by the World Health Organization (WHO), “disability” is an umbrella term for impairments, activity limitations, and participation restrictions, denoting the negative aspects of the interaction between an individual (with a health condition) and that individual’s contextual (environmental and personal) factors [25]. Disability is neither a simple biological nor a social phenomenon, and must be faced promptly and drastically, as mentioned in the “WHO global disability action plan 2014–2021: better health for all people with disability” [25]. Disability is a global public health issue because people with disability, throughout the course of their life, face widespread barriers to accessing health related services, such as rehabilitation, and have worse health outcomes than people without disability in high- as well as in low-income countries [25]. The action plan calls for member states to remove barriers and strengthen/extend rehabilitation, provide assistive devices, and support community-based rehabilitation. Accordingly, the action plan asks for the enhanced collection of relevant and internationally comparable data on disability research [25].

Adapted precision-exercise-based training programs (PEx) are emerging therapeutic options in pediatric oncology, and they fully encapsulate the directions given by the WHO’s global disability action plan [1,2,3,26,27,28]. PEx are run in hospitals as therapy prescribed by a sports medicine doctor teamed up with the CAYA-H’s pediatricians and performed under exercise scientists’ supervision [29]. The paradigm “exercise as medicine” is fulfilled by PEx, especially when it is started at the very onset of hematological malignancy and applied to the most medically fragile CAYA-H, including hematopoietic stem cell transplantation (HSCT) recipients [2,5,6,26,27,28]. PEx are inclusive in nature and can produce relevant and internationally comparable data by using a careful evaluation of the impact of exercise on physiological and social outcomes.

A new conceptualization of the correct phenotyping of populations before starting a PEx intervention was suggested by Scott et al. (2018), in order to facilitate personalized risk assessment and the development of targeted exercise prescriptions to optimally prevent or manage system toxicity after a cancer diagnosis [30]. Following this rationale, and because the efficiency of our systems is truly understood when the oxidative metabolism pathway is under stress (during an augmented metabolic request), we thought that introducing a performance test would fit with the next generation of precision-based exercise interventions [4]. An exhaustion test can contribute to the correct allocation into a specific phenogroup with a corresponding training protocol [30].

A meticulous approach to tailored exercise/sports programs includes a sensible functional evaluation of every CAYA-H to set the right amount of training, in terms of quality and quantity, exactly as pharmacological treatments are typically prescribed. This is a major challenge for clinical exercise physiologists and pediatricians; however, the reward could be significant, and the CAYA-H disability conversation flaws lessened. The collection of accurate data regarding the follow-up of CAYA-H’s exercise tolerance is not insignificant: due to the pharmacological toxicity related to life-saving treatments and the consequences of being bedridden, they experience a roller-coaster of symptoms that vary from days where PEx is fully manageable to days when severe fatigue is prevalent. The fatigue also affects a patient’s motivation to be evaluated and trained [31,32,33].

Customizing PEx requires a battery of functional evaluation tests, such as those used amongst athletes, including the evaluation of endurance, resistance, balance, and flexibility capacities. The complication is that functional evaluation tests must be performed in restrictive clinical settings and require a daily consultation between sport medicine experts and pediatricians. Evaluating an individual’s maximal exercise tolerance is the most accurate way to define PEx and to longitudinally assess their impact on CAYA-H [34,35]. Traditionally, the capacity of patients with chronic diseases, including cancer, has been evaluated using continuous exercise tests on ergometers and measuring the maximum aerobic capacity [4,34,35,36].

However, in order to check the activity profile of children and their usual activities, it seems that exclusively evaluating the oxidative metabolism chain during continuous exercise is not enough. A child’s activity consists of intermittent exercises such as jumps, turns, high-speed runs, and sprints, and resembles sports such as soccer and basketball. A Yo-Yo intermittent recovery test (Yo-Yo IRT) is commonly used to evaluate the performance of soccer players; it involves acceleration and deceleration phases interchanged with recoveries [37]. Through its use of incremental speeds and repeated shuttles of running, the test makes it possible to carry out intermittent exercise, leading to the maximal activation of the aerobic system, as well as determining an individual’s ability to recover from repeated exercise, with a high contribution from the anaerobic system [37]. CAYA-H are unable to use the original protocol suggested by Bangsbo et al. (2008), so we thought that an adapted version of the Yo-Yo IRT (Yo-Yo AD) would be a fair and safe approach to the assessment of CAYA-H’s individual performance and to evaluate the impact of PEx.

This study aimed at verifying whether the Yo-Yo AD test can provide a simple and valid way to obtain important information on an individual’s capacity to perform repeated, intense exercise and to examine changes in performance of CAYA-H performing precision-based exercise during cancer treatment. The present article deals with various physiological aspects of CAYA-H performing the Yo-Yo AD and the use of this test in a complex clinical setting. This study attempts to discuss the factors that are important when choosing an appropriate performance test for CAYA-H. Thus, the validity, reliability, and sensitivity of the Yo-Yo AD are addressed.

## 2. Materials and Methods

This was a single-center, analytic observational study, including both a cohort (CAYA-H, before and after exercise intervention) and a case-control (CAYA-H vs. healthy CTRL) design. Sample size calculation determined that a sample of 15 participants would be adequate to detect a difference of 25% in functional ability (6MWT or TUDS) between the participants, as opposed to healthy children, with a power of 0.80 (α = 0.05). Every attempt was made to avoid unnecessary discomfort and disturbance to CAYA-H and parents according to the ERICE statement about the cure and care of long-term survivors of childhood cancer [38]. All children and parents provided written consent to participate in the study that was approved by the University of Milano Bicocca Ethics Committee (registered number 2017/284). Personal data was treated in compliance with the European standard principles of confidentiality (n. 2016/679). The protocol was registered at ClinicalTrials.gov PRS (n. NCT04090268).

### 2.1. Participants and Intervention

Eligible CAYA-H were: (1) those treated for any hematological malignancy (including relapses) at the Maria Letizia Verga center (Monza, Italy) from April 2017 to July 2020, with the participants being aged 7 to 19 years old; (2) CAYA-H within 4 weeks from diagnosis, as well as those in the last weeks of treatment before being off-therapy. A small group of patients in their late post-HSCT course, with respiratory or joint grafts versus host disease and/or osteonecrosis were also included.Another criterion was that each CAYA-H agreed to participate in a round of an 11-week combined training program (endurance, resistance, balance, and flexibility) three times per week that was individually targeted, with workloads ranging from moderate to intense exercises, depending on the daily clinical conditions. A control group of 18 healthy children (CTRL) of matching age and gender was evaluated as they took part in recreational (non competitive) summer-time activities.

The main reason for CAYA-H not participating in the study was logistical (CAYA-H living outside of province or region). A small number of families refused to attend the PEx sessions because of a lack of interest in participating in the research project. Each participant was informed by the referring pediatric hematologist of the possibility of participating in the research protocol. Subsequently, a sports medical doctor evaluated the possible risks of performing precision training. 

CAYA-H were phenotyped according to their clinical history and to the intensity of their treatment protocol, by using the Intensity of Treatment Rating (ITR-3) [39]. Each CAYA-H was allocated in one of the following PEx protocols (Table 1): moderately intensive treatment, very intensive treatment, and most intensive treatment. CAYA-H in their late post-HSCT course, with respiratory or joint graft versus host disease and/or osteonecrosis, adhered to the moderately intensive PEx protocol.

Every day of the training session, although specific workloads were settled according to the functional evaluation performed before the start of the PEx, a consultation between the pediatrician and the sports medical doctor led to the avoidance of exercises considered as dangerous due to the clinical history of the patient. The daily training was changed mostly in type and intensity if a specific intercurrent clinical condition was known. Two possible examples are as follows: (1) in the case of severe anemia, the cardiorespiratory exercise was reduced or avoided; (2) in the presence of suspected lower limb osteonecrosis a continuous maximum amount of cardiorespiratory exercise (2–6 min) was refrained from in order to preserve the joint structure.

### 2.2. Assessment of the Exercise Tolerance

Figure 1 shows the Yo-Yo AD execution.. The number of shuttles for each speed, and the cumulative distance are presented in Appendix A section. The speeds ranged from easy walking (3.0 km/h) to low race pace (8.0 km/h).

The Yo-Yo AD involved the repetition of several shuttles, each comprising a 20-m route with a round trip, at pre-established speeds, and each followed by 10 s of rest. The shuttles were time-marked by acoustic sounds on an audio track, signaling the start of each new shuttle and the recovery time between one shuttle and the next., (e.g., double acoustic sound: go for 20 m; wait until the acoustic sound; back for 20 m; acoustic sound: recovery phase; double acoustic sound: start of a new shuttle). Each speed change was indicated by a vocal recording, clearly stating the new speed (e.g., “You are now running at 4 km/h”). If a CAYA-H did not finish the shuttle within the sounds indicating the start of the next distance twice in succession, the test was considered as completed due to exhaustion and the best individual performance was the total distance covered by the CAYA-H.

Heart rate (HR) and oxygen saturation (SpO_2_) were continuously monitored by a pulse oximeter (OxyTrue A, bluepoint MEDICAL GmbH & Co., Selmsdorf, Germany) and a correlation between the HRs and times was evaluated. The slope of HR vs. time was considered as an individual’s walking/running ability [40]. At the basal evaluation (T0), soon after the recruitment to PEx, the test was performed twice on two different days in the same week; the first attempt was considered as a familiarization session, while the second was the effective evaluation. No verbal encouragement, music, or feedback was allowed during the test sessions. 

The clinical conditions of each CAYA-H were evaluated before the execution of the Yo-Yo AD by a pediatrician and a sports medicine doctor; the latter was also in charge of supervising the full evaluation session.

In order to test the reliability of the Yo-Yo AD, 15 CAYA-H repeated the test 7–10 days after the execution of the previous measure.

### 2.3. Assessment of Functional Capacity and Strength

The Timed Up and Down Stairs test (TUDS) was performed to measure the general neuro-muscular aspects of dynamic posture [41]. The 6-min walking test (6MWT) was performed to assess the system response to moderately aerobic exercises to intensive performance requiring lactic acid anaerobic pathway (Silverline 868793 Mini measuring wheel) [42]. The strength of both quadriceps was investigated through the execution of a five maximum repetitions test (5RM) [43]. A leg extension machine was modified to fit the body dimensions of CAYA-H of different ages (leg extension Alpha Pro, multi-function bench, Kettler, Ense-Parsit, Germany).

### 2.4. Statistical Analysis

Values were expressed as mean (±standard deviation). D’Agostino and Pearson’s omnibus normality test was used to check whether the values were from a Gaussian distribution. Pearson’s correlation test was used to assess the reliability (test-retest) of performance during the Yo-Yo AD [44]. The ROC curve was used to evaluate the sensitivity and specificity of the Yo-Yo AD [45]. The statistical significance of the difference between mean values, considering the different circumstances, was evaluated as follows: (1) In case of differences between CAYA-H at T0, T1 vs. CTRL, an ordinary one-way ANOVA, followed by a Kruskal–Wallis test, with a Dunn’s multiple comparative test; the same applied to different groups of CAYA-H (male vs. female, or children vs. teens) at different time points (T0 vs. T1). (2) In case of differences between CAYA-H at T0 vs. T1, a Wilcoxon matched-pairs rank test was used. The regression analyses were performed using the r^2^ method. If there was no statistically significant difference between the two regression lines, the unified line equation (slope and pooled intercept) was used. The level of significance was set at *p* < 0.05. All statistical analyses were performed using a commercially available software package (Prism 8.4.3: GraphPad, La Jolla, CA, USA)

## 3. Results

Two-hundred-five CAYA-H were eligible (Figure 2) and of the 213 aged ≥ 7 years who attended PEx, 40 dropped out (<15% of adherence to the PEx sessions) due to lack of compliance, such as living too far away from the hospital or a busy family schedule when young siblings were in the same familiar nucleus. Ninety-seven CAYA-H performed the full battery of tests and participated in the basal (T0) and post-PEx (T1) evaluation; only their data are presented in this paper (Figure 2). Their adherence to PEx was >64% of the total amount of the training sessions (33 total sessions).

The reasons for ending the Yo-Yo AD were: 43% did not finish the shuttle in the established time, 33% voluntarily finished for reported unbearable fatigue, and 24% finished due to chest or leg pain. No major adverse events (falls, syncope, muscle strains, post-exercise asthma) occurred during or after the test sessions. Very medically fragile CAYA-H fully recovered in 48 h from the test, when muscular fatigue ceased completely.

The clinical characteristics of the participants and the intensity of their treatment protocol are shown in Table 2. Table 2 also shows the CAYA-H that were long term follow-up patients (>5 years after their HSCT) but with impaired functional walking ability due to graft versus host disease and/or osteonecrosis.

**Table 2 cancers-14-01187-t002:** Clinical characteristics of children, adolescents, and young adults with hematological malignancies and intensity-of-treatment rating.

Clinical Characteristics	Num, (%)	Treatment Protocols
Patients, total	97	
Aged 7 < x < 11 years	58, (60%)	
Aged 11 ≤ x < 22 years	39, (40%)	
*Level 4: Most Intensive Treatments*	25, (26%)	
Relapsed Disease—Excluding Hodgkin Lymphoma, first relapse	8, (8%)	IntReALL SR or Personalized treatment
HSCT—All diseases	12, (13%)	Personalized treatment
AML	5, (5%)	AML 2013/01
*Level 3: Very Intensive Treatments, Total*	16, (17%)	
Relapse Protocols for Hodgkins	3, (3%)	Personalized treatment
ALL (High Risk, Very High Risk, T-cell)	10, (11%)	AIEOP BFM ALL 2009
HL (Stages 3B or 4/High Risk)	1, (1%)	EuroNet-PHL-C2
NHL (Group C or Stage 4)	2, (2%)	Euro LB-02/ NHL97
*Level 2: Moderately Intensive Treatments, Total*	51, (53%)	
ALL (Low, Standard, or Intermediate Risk; precursor B cell)	36, (37%)	AIEOP BFM ALL 2009
HL (Low/Intermediate risk: all stages except IIIB, IVB)	13, (14%)	EuroNet-PHL-C2
NHL (Stages 1, 2, 3 and Groups A, B)	2, (2%)	Euro LB-02/ NHL97
*Level 0: Off Therapy, Total*		
*Long-term HSCT, with GvHD and/or ON*	5, (4%)	Personalized treatment

ALL: acute lymphoblastic leukemia; AML: acute myeloid leukemia; HL: Hodgkin lymphoma; NHL: non-Hodgkin HSCT: hematopoietic stem-cell transplant; GvHD: graft versus host disease; ON: osteonecrosis.

Twenty-three CAYA-H (24%) performed PEx for 2–4 rounds consecutively, due to severe chronic functional impairment and/or cardiorespiratory system inefficiency at the end of one round of 11 weeks of training.

### 3.1. Reliability

The test-retest reliability showed the consistency of the Yo-Yo AD measures over time (r2 0.9764; *p* < 0.0001) and is represented in Figure 3A. The Yo-Yo AD was conducted on two different days (in 72 h), on a sub-group of 14 CAYA-H, by the same expert operator.

Table 3 represents the correlations at T0 and T1 between the Yo-Yo AD and the functional tests (6MWT average 496.0 ± 122.0 m, TUDS average 8.05 ± 4.33 s, Leg Extension average 36 ± 18 kg). 

### 3.2. Sensitivity

The sensitivity of the Yo-Yo AD measures, assessed by the ROC curve, is represented in Figure 3B. The area below the curve (AUC) was 0.7 and the confidence interval at 95% was 0.6208–0.7702. A cut-off performance for medically fragile CAYA-H was settled at 740 m (sensitivity: 36.99%; specificity: 82.47%) and for well-performing CAYA-H at 1420 m (sensitivity: 82.61%; specificity: 43.30%).

### 3.3. Validity

Figure 4 shows the impact of PEx on the performance of the Yo-Yo AD. There was a significant improvement in the performance at T1 when compared to T0 (T0 = 946.6 ± 438.2 vs. T1 = 1352.3 ± 600.6 m, *p* < 0.001). CAYA-H had, on average, a lower performance than CTRL (1754.0 ± 444.0 m) both at T0 (*p* < 0.001) and T1 (*p* = 0.013).

The correlation between time and HR during the Yo-Yo AD execution at T0, i.e., the efficiency of the individual walking/running, is shown in Figure 5A. Three different patterns were identified based on the cut-offs obtained by the ROC curve analysis (<740 m, 740 < x < 1460 m and >1460 m, respectively). The three different patterns represent satisfactory, intermediate, and poor performances, depending on the HR response to the increasing workload (slopes 3.275; 2.189; 1.624, respectively). The slopes and intercepts of these three patterns were statistically different (*p* < 0.001).

The three patterns after PEx (T1) are shown in Figure 5B. The significant decrease in the averaged HR/Time slopes after 11 weeks of PEx suggests the beneficial effect of the PEx on the heart-rate response to the exercise (T0 = 2.175 ± 0.844 vs. T1 = 1.732 ± 0.891 slope, *p* < 0.016). There was a statistically significant reduced slope (*p* < 0.001) for the satisfactory and intermediate patterns (slopes 1.722 and 1.464, respectively), when compared to T0, but not for the poor performances (slope 3.858). The latter was significantly worse than T0 (*p* < 0.001).

When comparing the Yo-Yo AD performances according to sex in CAYA-H, there was no statistically significant difference between the groups at T0 (male 990.2 ± 585.7 vs. female 1153.0 ± 485.0 m, *p* = 0.999) or at T1 (1339.0 ± 739.4 vs. 1386.0 ± 588.1 m, *p* = 0.999). The comparison between the performances according to cohorts of age at T0 and T1 showed a statistically significant difference between the groups at T0 (teen 1251.0 ± 803.4 vs. prepuberal children 803.4 ± 408.3 m, *p* = 0.008), but not at T1 (teen 1449.0 ± 636.8 vs. prepuberal children 1099.0 ± 693.5 m, *p* = 0.127).

## 4. Discussion

This study attempted to prove that the Yo-Yo AD can be safe and reliable when evaluating the physiological aspects of exercise performance in CAYA-H with varying degrees of fragility, including the most medically fragile HSCT recipients. We also tried to confirm that the Yo-Yo AD could evaluate the effect of PEx on performance. PEx is not detrimental to CAYA-H and has a beneficial effect on their exercise tolerance, as already reported by previous studies [1,2,3,27,28,34,35,36,43].

The risk of falling, parental motivation and fears, clinical conditions, and cultural prejudice on the part of family and health-care professionals (“exercise for CAYA-H is not a priority”; “CAYA-H are medically fragile children, not athletes”; “CAYA-H can’t withstand stairs”; “CAYA-H can’t train outdoors”; “my child is tired, she/he can’t exercise today”; “when on oxygen you can’t train CAYA-H”), generate a multifaceted scenario that must be taken into account when a PEx is hosted in one hospital.

The application of precision-exercise-based training programs for medically fragile children, not only in oncological settings, requires a cultural change from the diagnosis and treatment of illness to lead to improvements in health [31,32,33]. PEx have a preventive function against the possible effects of cancer or oncological treatment on skeletal muscle, cardiorespiratory system, and bone tissue. Their benefits extend beyond a simple improvement in exercise capacity and include improved feelings of physical self-perception and satisfaction with life. PEx works on intensity, volume, frequency, and recovery, and optimizes individual physiological and psychological adaptations [1,2,3,36,43]. PEx is the new frontier in clinical exercise physiology, helping to induce more efficient oxidative metabolism (i.e., the main system of energy supply within cells) and to boost the adaptive response of bones in vulnerable patients. The heterogeneity in response creates the strong hypothesis that a precision-oncology approach is required to optimize the benefits and safety of exercise as a candidate antitumoral strategy [46].

The use of generic exercise prescriptions may actually be masking the full therapeutic potential of exercise treatment in the oncological setting. Essentially, the manipulation of training variables, such as volume, intensity, frequency, and recovery is an attempt to systematically structure training through phases to optimize physiological and psychological adaptations in an athlete that is also a patient with cancer. Finally, work recovery and rest are fundamental to restore the availability of nutrients and energy substrates to replace the components needed by the systems (proteins in the muscle) [30].

### 4.1. A Safe Test

The Yo-Yo AD is an exhaustion test that could enhance the collection of relevant and internationally comparable data about the impact of PEx on CAYA-H with disabilities, as recommended by the WHO’s global disability action plan (2014–2021) [25].

Testing exhaustion in CAYA-H can be intimidating for both patients and investigators. Most clinicians have a weak understanding of exercise physiology in children with cancer and the benefits of exercise in CAYA-H are mostly unknown. Currently, there is a growing body of knowledge about the preventive effects of exercise in clinical pediatric settings [32,33], but a cultural change from the treatment of illness to the recovery of health in CAYA-H survivors is long overdue [31]. According to the American College of Sports Medicine recommendation, when PEx and, consequently the evaluation of a basal performance, are necessary, the balance between the increasingly recognized health benefits of even low levels of physical activity and the rarity of cardiovascular events provoked among those with established cardiovascular disease needs to be taken into account [47]. A medical doctor was always present during Yo-Yo AD testing and the clinical history of each CAYA-H was investigated before the test: structural cardiovascular abnormalities, most notably, hypertrophic cardiomyopathy, were identified before starting the exhaustion test. Moreover, in order to provide the best help in case of a cardiovascular emergency, all the sanitary and nonmedical personnel attended pediatric basic life-support training in order to provide immediate assistance, such as dialing 112, initiating bystander cardiopulmonary resuscitation, and using an automated external defibrillator [47]. HR and SaO_2_ were continuously monitored in order to verify whether any undesirable tissue hypoxia events occurred during the test. With these precautions, the Yo-Yo AD was also used for CAYA-H with complex clinical conditions, including osteonecrosis, mild restrictive pulmonary disease, reduced cardiac output, myopathies, and neuropathy.

The Yo-Yo AD was a challenge for each CAYA-H, but an easy “exit strategy” or contingency plan was granted to each child and adolescent by letting them know that they could stop whenever they wanted. 

A significant amount of time was spent in training the exercise scientists to report every testing and training session performed by each patient, on a daily basis. We considered relevant adverse effects as possible outcomes of interest because we recognize that adverse-effects data are often handled with less rigor than the primary beneficial outcomes of a study [46]. Because only reversible minor adverse effects were noted during this study, we considered PEx as being successfully introduced in a complex clinical setting. When a possible sign or symptom of systemic inefficiency was found (e.g., desaturation during the exhaustive test due to a pre-clinical pulmonary disease), we promptly referred the child to her/his pediatrician.

### 4.2. A Reliable and Valid Test

We tried to demonstrate that the Yo-Yo AD is a reliable test for CAYA-H, both in terms of test-retest reliability and of internal consistency [44,45]. We did not measure the inter-rater reliability; in fact, all the tests were performed by the same two operators.

The test-retest reliability was not performed in the CTRL because the healthy adolescents and children, as expected, worked at sub-maximal workloads until the end of the test (2040 m). Healthy peers can easily perform the heavy workloads of the original Yo-Yo IRT in order to evaluate their maximal aerobic capacity [37]. In the CTRL, the Yo-Yo AD gave us an indication of the participants’ walking/running ability during sub-maximal workloads, and the same applied to the highest-performing CAYA-H.

According to Currell et al. (2008), there are different ways to attest the validity of performance protocols: (i) logical validity; (ii) criterion validity; and (iii) construct validity [44]. It can be argued that sports performance is a construct. Construct validity refers to the degree in which a protocol measures a hypothetical construct, in this case, performance. It can be measured by comparing two different groups of participants with different abilities. In our case, we compared CTRL healthy children with our CAYA-H and CAYA-H with different clinical characteristics, from medically fragile to normally performant. A test with good construct validity can easily highlight the differences in performance between different groups, as the Yo-Yo AD did.

### 4.3. Exercise Tolerance, Strength, and Metabolic Pathways

In our study, the 6MWT evaluated the individuals’ physical functioning, but we are inclined to assume that, indirectly, it could also give information about individuals’ ability to produce energy through two different metabolic pathways: the mitochondrial respiration and anaerobic glycolysis. The efficiency of energy production could range from very good to severely compromised. This is new information that we derived from the fact that some participants could perform the test without showing any sign of exhaustion, while others were notably above their respiratory threshold. The same applied for the TUDS, where the efficiency of two other metabolic pathways (phosphocreatine hydrolysis and anaerobic glycolysis) was mainly presumed to have been used to perform exercise lasting a few seconds [37]: the majority of our participants took less than 10 s to perform the exercise, while the most medically fragile children needed more than 25 s. The correlations between 6MWT or TUDS and Yo-Yo AD performances show that CAYA-H have varying degrees of ability to use all the integrated metabolic pathways. For the most medically fragile CAYA-H, the three pathways are inefficient at satisfying the energy requirements of the skeletal muscles, and their final performance is poor. 

Our CAYA-H were trained using methods aimed at increasing both skeletal muscle strength and endurance. The relation between the strength of the quadriceps, measured directly on a leg extension machine, and the Yo-Yo AD seems to confirm that neuromuscular performance during endurance exercise also plays an important role in CAYA-H [48]. Strength training carried out in conjunction with endurance training could have had a beneficial effect on the neuromuscular characteristics of the CAYA-H, as already shown both in athletes and in sedentary people [48].

### 4.4. Patterns of Efficiency

Three possible patterns of HR response were found in the CAYA-H. The highest-performing CAYA-H were able to perform better during the Yo-Yo AD test, while keeping their HR response lower. This could suggest better muscle tolerance/quality (i.e., better mitochondrial respiration, etc.). In athletes, the levels of muscle phosphocreatine are higher, and the concentration of lactates is low at the end of the Yo-Yo IR1 [37]. Although the Yo-Yo AD test was adapted to allow all the participants to perform it, some CAYA-H presented a mild-to-severe inefficiency in their oxidative metabolism pathways [4]. In case of severe atrophy, the stores of muscle glycogen can be very low, and the rate of glycolysis can be more pronounced than when the aerobic component is still efficient [8]. For these CAYA-H, the test can be comparable to the Yo-Yo IR2 of Bangsbo et al., where the efficiency of a healthy cardiopulmonary system is deeply strained. Our fragile CAYA-H performed the 6MWT as an exhaustion test, as already noted by Geiger et al. [42].

The area under the ROC curve emphasized how the Yo-Yo AD can identify the most fragile CAYA-H (who could not pass the cut-off distance of 740 m) and the highest-performing (who did not perform less than 1420 m). The ROC curve was built considering the average distance reached at T1 as the normal value to be achieved compared to that at T0, so the Yo-Yo AD could evaluate the effect of the PEx. The fact that the 95% confidence interval did not include 0.5 (representative of the diagnostic indifference threshold) is indicative of good power discrimination by the test, regarding the possibility of execution, both in performing patients and in fragile patients.

### 4.5. Effect of PEx

The Yo-Yo AD showed that PEx brought CAYA-H performance closer to that of CTRL, although there was a significant difference between the groups. However, at T1, a group of a few CAYA-H showed poor cardiopulmonary efficiency and did not consistently improve their performance, despite the PEx. The multiple complications experienced by these patients are the likely explanations. At baseline, the clinical conditions of some of the CAYA-H prevented their execution of the Yo-Yo AD. After 11 weeks of PEx, the test was carried out, but the cardiopulmonary efficiency of these participants was still characteristic of fragility. Other CAYA-H were diagnosed with relapse or pulmonary and/or urinary infections during their PEx and shifted towards a more intensive drug treatment, which led to long-lasting bed rest. These CAYA-H typically shifted from good cardiopulmonary efficiency towards inefficiency and fragility.

In order to perform PEx, we had to face the cultural prejudice of family and health-care professionals. A multidisciplinary team was needed in order to face any consequences of PEx, such as the risk of falling, as well as to support parental motivation and counteract any fears related to the participants’ fragile clinical conditions. The more frequent reasons raised by parents for avoiding the training sessions were: “exercise in CAYA-H is not a priority”; “CAYA-H are medically fragile children, not athletes”; “CAYA-H can’t withstand stairs”; “CAYA-H can’t train outdoors”; “my child is tired, she/he can’t exercise today”; “when on oxygen you can’t train CAYA-H”. Once the multidisciplinary team consulted with parents, many fears were faced, and training was finally considered as an opportunity to physically and psychologically benefit each child.

### 4.6. Role of Sex and Age in Performance

According to Bangsbo’s study [37], post-pubertal males performed better when compared to females. This was not the case in CAYA-H, who showed a similar performance: cancer treatment and muscle disuse seemed to inhibit the effect of testosterone on skeletal muscle strength, limiting the effect of male sex during the Yo-Yo AD. Overall, the male CAYA-H may have had a higher degree of fragility than the females, which was not so problematic whenwhen they performed exercises that required intermittent phases of activity. After 11 weeks of PEx, there were no differences related to sex, but in this case, there was also a limit imposed by the plateau effect of the test itself.

In line with Bangsbo’s findings [37], for our CAYA-H, age also influenced performance because the teens performed better than the younger boys and girls. The lack of difference between the performance of the two groups at T1 may be attributable, again, to the plateau effect of the test.

### 4.7. Limitation of the Study

The limitations of this study were the lack of possible criteria for the randomization of the participants in the intervention and control groups. We are aware of the fact that a randomized control trial would have provided the highest evidence level for this experimental intervention. We considered having a CTRL group of CAYA, but upon a careful evaluation, the use of two truly homogenous groups appeared to be impossible. We accept that our study has a lower ranking in the hierarchy of evidence as losing the power of randomization causes the study to be more susceptible to bias and confounding.

In regard to the inter-rater reliability, we plan to evaluate, in a further study, the consistency of measurements performed by different operators, with various degrees of experience, in running performance tests in children with cancer.

## 5. Conclusions

Yo-Yo AD was shown to be a valid tool to evaluate and follow up individuals’ capacity to perform repeated, intense exercise and to examine changes in the performance of CAYA-H attending a precision-based exercise program during cancer treatment. All the CAYA-H, including the most medically fragile, could safely perform the Yo-Yo AD during their hospitalization. After 11 weeks of precision exercise training, the majority of the CAYA-H could resume regular physical activity, including running at high speed and sprinting. This is a preliminary transition toward their return to sports once they are back to their communities, after the intensive phases of cancer treatment.

## Figures and Tables

**Figure 1 cancers-14-01187-f001:**
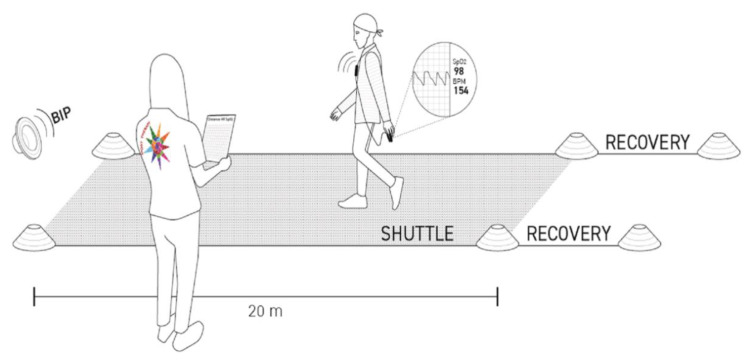
Schematic representation of the Yo-Yo AD test execution. One exercise scientist monitored the safety and correct execution of the test. The vital parameters, such as heart rate and oxygen saturation, were recorded during the test. An acoustic sound signaled the start of the shuttles and recovery phases.

**Figure 2 cancers-14-01187-f002:**
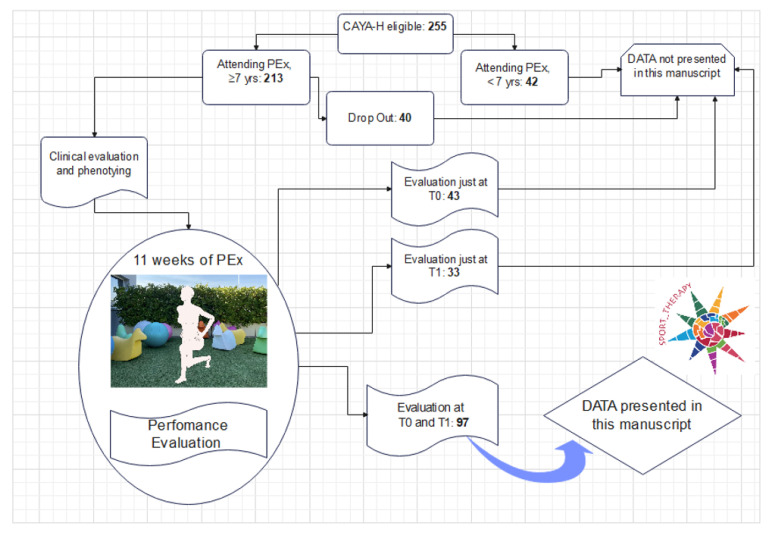
Children, adolescents, and young adults with hematological malignancies (CAYA-H) flow chart, from eligibility to precision exercise training program (PEx) participation. The flow shows the Yo-Yo AD test evaluations considered as data presented in our manuscript.

**Figure 3 cancers-14-01187-f003:**
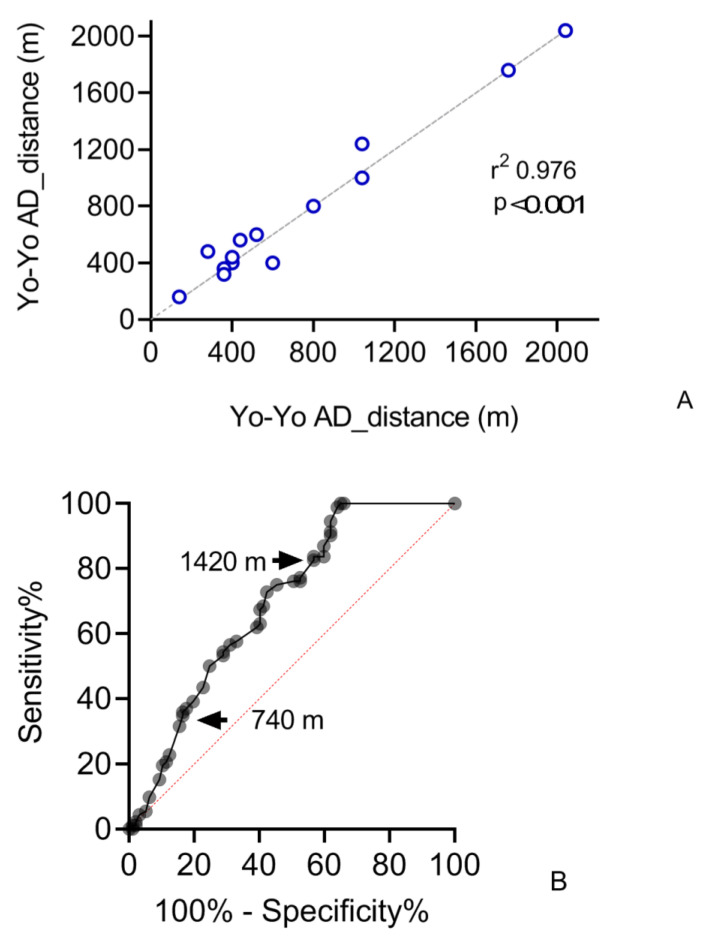
(**A**). Test-retest reliability of distances performed during the intermittent recovery Yo-Yo adapted test (Yo-Yo AD) of a sub-group of children, adolescents, and young adults with hematological malignancies. Almost all CTRL (93%) reached the plateau of the Yo-Yo AD, i.e., the maximum distance (2040 m in 28 min). CAYA-H did not reach the plateau at T0 and T1 in 84% and 66% of cases, respectively. (**B**). Sensitivity of the Yo-Yo AD measures, assessed by the receiver operating curve (ROC). A cut-off performance for medically fragile CAYA-H was settled at 740 m (sensitivity: 36.99%; specificity: 82.47%) and for high-performing CAYA-H at 1420 m (sensitivity: 82.61%; specificity: 43.30%.).

**Figure 4 cancers-14-01187-f004:**
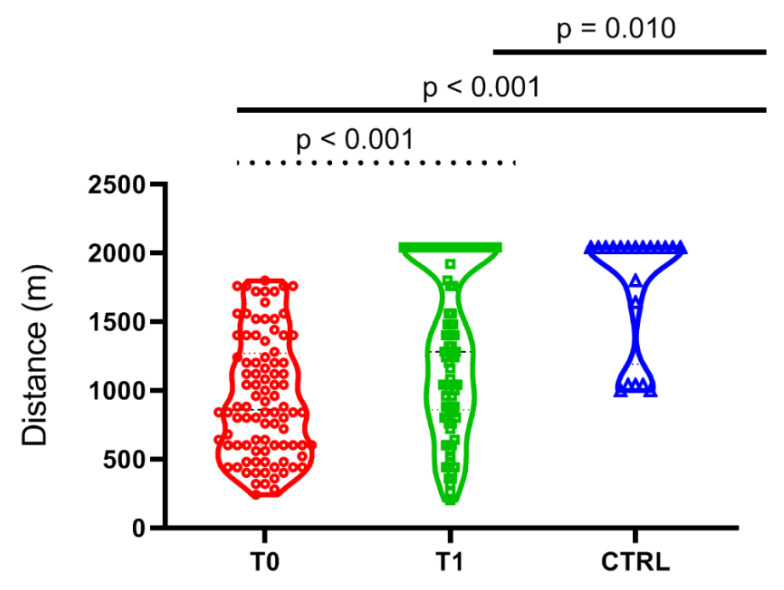
Violin plots showing the distribution of the children, adolescents, and young adults with hematological malignancies (CAYA-H) according to the total distance performed during a Yo-Yo AD. A significant difference was observed between the average scores obtained at the two different time points (T0, basal value vs. T1, after 11 weeks of precision based exercise program). The statistical significance between CAYA-H (dotted line, matched-paired test) and groups of healthy pairs (CTRL) is represented (continuous lines, ANOVA).

**Figure 5 cancers-14-01187-f005:**
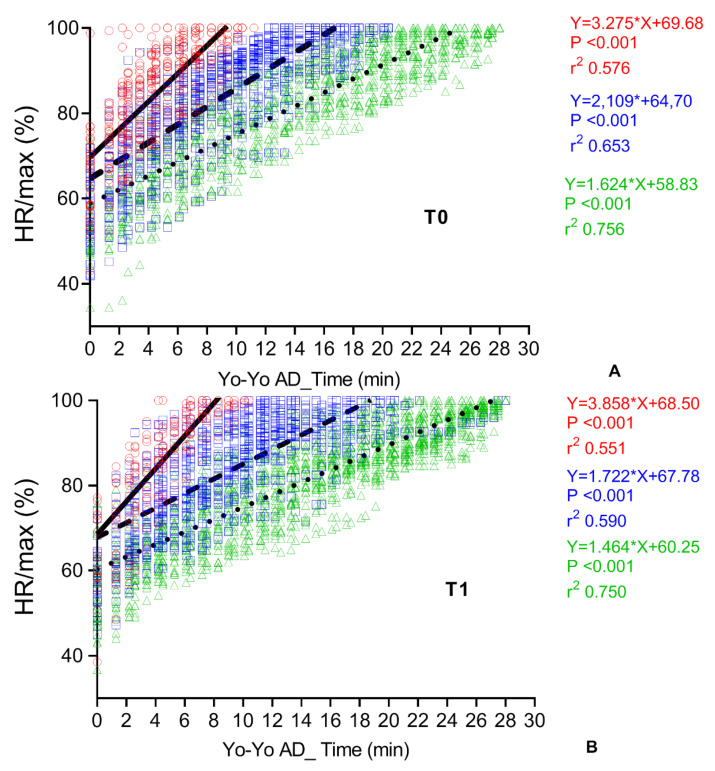
(**A**,**B**). Individual and averaged correlation between time and heart rate (HR) during the Yo-Yo AD execution at the basal evaluation (T0) and after 11 weeks of precision-based exercise program (T1). HR is expressed as a percentage of maximal heart rate calculated during the test session. Three patterns identified by different slopes represent satisfactory (triangles, dotted line), intermediate (squares, dashed line), and poor performances (circles, continuous line). See text for further explanation.

**Table 1 cancers-14-01187-t001:** Precision exercise training protocols according to the clinical phenotyping of children, adolescents, and young adults with cancer (see also Table 2 for the characteristics considered to identify the intensity of cancer treatment). Type, frequency, pattern, and progression are common to all protocols. The intensity, time, and volume of exercise is different for most intensive, very intensive, and moderately intensive cancer treatment.

Type	Cardiorespiratory	Resistance	Neuromotor	Flexibility
Elaborated meaning	Regular, purposeful exercise that involves major muscle groups and is continuous and rhythmic in nature (e.g., walking on an elastic trampoline)	Major muscle groups with a variety of exercise equipment and/or body weight (e.g., leg extension machine). Different muscle groups are used every day	Exercise involving balance, agility, coordination, and gait, proprioceptive training, and multifaceted activities (e.g., balance on proprioceptive surfaces)	A series of flexibility exercises for each of the major muscle-tendon units
Frequency	≥3 day × week^−1^	≥3 day × week^−1^	≥3 day × week^−1^	≥3 day × week^−1^
Pattern	One continuous session of supervised exercise per day with active recoveries for each 2 min of activity	Rest intervals of 1 min between each set of repetitions	/	At the end of each training session. One repetition
Progression	Increased exercise volume each 4 weeks	Increased exercise volume each 4 weeks	/	/
**Most intensive treatment, exercise training protocol**				
Intensity	Light–moderate, 40–60% of HRR	Light, 40–50% of the 1 RM	Indeterminable	Stretch to the point of feeling tightness or slight discomfort
Time	30 min × week^−1^	30 min × week^−1^	15 min × week^−1^	15 min × week^−1^
Volume/repetitions	≥120 MET min × week^−1^	8–12 repetitions, 2–3 sets	/	30 s
**Very intensive treatment, exercise training protocol**				
Intensity	Moderate–vigorous, 60–80% of HRR	Moderate–hard, 60–70% of the 1 RM	Indeterminable	Stretch to the point of feeling tightness or slight discomfort
Time	60 min × week^−1^	60 min × week^−1^	30 min × week^−1^	30 min × week^−1^
Volume/repetitions	≥240 MET min × week^−1^	8–12 repetitions, 3–4 sets	/	30 s
**Moderately intensive treatment, exercise training protocol**				
Intensity	Vigorous, 70–90% of HRR	Hard, 70–80% of the 1 RM	Indeterminable	Stretch to the point of feeling tightness or slight discomfort
Time	60–70 min × week^−1^	60–70 min × week^−1^	30–35 min × week^−1^	30–35 min × week^−1^
Volume/repetitions	≥240 MET min × week^−1^	8–12 repetitions, 3–4 sets	/	30 s

HRR: Heart-rate reserve (HR at rest and maximum measured during an exhaustion test); MET: metabolic equivalent; 1 RM: one-repetition maximum.

**Table 3 cancers-14-01187-t003:** Correlations between Yo-Yo AD and other performances: 6-min walking test (6MWT), Timed Up and Down Stairs (TUDS), quadriceps strength (leg extension).

**6MWT vs. Yo-Yo AD**		
	T0	T1
Equation	Y = 4.046 × X − 1.147	Y = 4.583 × X − 1.278
*p* value	<0.001	<0.001
R square	0.420	0.554
**TUDS vs. Yo-Yo AD**		
	T0	T1
Equation	Y = −162.8 × X + 2.210	Y = −249.5 × X + 2.970
*p* value	<0.001	<0.001
R square	0.215	0.377
**LEG EXTENSION vs. Yo-Yo AD**		
	T0	T1
Equation	Y = 18.31 × X + 381.3	Y = 22.86 × X + 367.8
*p* value	<0.001	<0.001
R square	0.417	0.335

## Data Availability

The data presented in this study are available on reasonable request from the corresponding author. The data are not publicly available due to sensitive information regarding the clinical status of underage patients.

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
