# Peer review of "The Impact of a Precision-Based Exercise Intervention in Childhood Hematological Malignancies Evaluated by an Adapted Yo-Yo Intermittent Recovery Test"

_cancers, 2022, doi:10.3390/cancers14051187_

Round 1
Reviewer 1 Report
One primary question is whether these study participants received any chemotherapy or other necessary treatments for cancer before and during the re-intervention period. If so, what types of chemotherapeutic agents were prescribed. For the purposes of determining the precise exercise prescriptions to be given to cancer patients, it may be important to know whether they received specific chemotherapy during or before (how long?) the intervention. Whether or not specific chemotherapy is received may affect the effectiveness of the exercise intervention, and this information should be detailed again in the subject's profile. This should be critical to the data interpretation, please address it.
Please consider these comments and resubmitted with new revision.
Reviewer 2 Report
Dear authors, thank you for the point-to-point answers and the clear revision of this manuscript. I think, the manuscript was improved a lot. But still, some aspects are, from my point of view, need to be revised (minor revision).
Introduction: The introduction was improved and some references were added. From my point of view, there are several sentences and expressions that must be supported by a citation. If this is the authors' own opinion, this should not be stated in the introduction, but in the discussion section.
Page 2 line 58-61: "Furthermore, CAYA-H fate is to become... " Reference needed!
Page 2 line 64-68: "CAYA-H and their families all over the world..." + "As a consequence, their prospects to find..." Reference needed!
Page 2 line 76-77: "Each CAYA-H will experience disability..." Reference needed!
Page 2 line 89-91: "Accordingly, the action plan asks..." Reference needed!
Page 2 line 97- page 3 line 102: "The paradigm... PEx are inclusive in nature and can produce relevant and internationally comparable data by using a careful evaluation of the impact of exercise on physiological and social outcomes." Reference needed!
Page 3 line 102 (whole paragraph): Reference needed!
Page 3 line 131-133: "But in order to check..." Reference needed!
Methods and Materials: This section improved as well, but some parts should be transferred into the results section, from my point of view.
Page 4 line 181-183: These are results, not methods?
Page 6 line 236-243: These are results, not methods?
Discussion: The discussion section was revised intensively. Some paragraphs are good, but some really miss out on references:
Page 15 paragraph 4.3 Exercise tolerance, strength and metabolic pathways - there's only one single reference, that's not enough, from my point of view. E.g., the first sentence needs to be supported by a reference or is this new information that was investigated in this study? Please revise this paragraph.
Page 15 paragraph 4.5: There's not a single reference in the whole paragraph. That's really a problem. Please revise this paragrah and add current references to compare/support/discuss your findings.
All in all, the manuscript really improved in this revised version. But some aspects need to be improved still, from my point of view, I'm sorry.
All the best for the authors!
Reviewer 3 Report
No further edits from me. Great job.
Author Response
We thank the reviewer for her/his appreciation.
This manuscript is a resubmission of an earlier submission. The following is a list of the peer review reports and author responses from that submission.
Round 1
Reviewer 1 Report
This is an very interesting and important investigation. The authors raised an important issue regarding to the various physiological aspects in CAYA-haem performing the Yo-Yo AD and with the use of this test in a complex clinical setting and then look at how the effect of PEx can be measured. Overall, it is an well written manuscript in both the rationale statement in the introduction and data presentation. However, there were still several information missed, and it would be great if these information could be added in the revised version. (1) The more detailed descriptions for the participants should be stated in either method or results (i.e. their basic characteristics in cancer stage, the fundamental hemotological test data, etc.), which would be important for readers to classified for applying the knowledge form this study to their clinical practice, although the brief inclusion criteria had been mentioned in the participants; (2) A detail experimental flow chart for the timeline or procedure would be helpful if it is added; (3) Are there any study limitations or precautions for this adopted YoYo test? There were no specific paragraph stated about these issues in the discussion, please add. I will consider the revised version if the above pointed parts are added.
Reviewer 2 Report
Authors have attempted to address precision-based training programs. However, the main idea conveyed throughout the manuscript is mostly the validity or reliability of Yoyo AD, how it was used by the investigators, rather than any precision-based exercise training. In particular, main results do not really show any “impact of precision exercise intervention” While authors have emphasized that children, adolescents and young adults with haematological malignancies, study participants were 10.58 ± 4.5 years old which dose not support the authors’ point. Further, while authors reported this is a valid test, the validity may not be established in the absence of direct comparison with the gold standard, although reliability has been investigated. There needs more clarification on the statistical analysis as well. Please see below for specific comments.
Simple summary
- Line 15: Please specify what “disability” is meant to be described, like physical social, psychological, and etc in the summary and throughout the manuscript.
- Line 21: this sentence is more likely imply that this manuscript is about validity of YO YO AD, rather than the impact of precision exercise intervention thus it may give confusion to some extent.
Abstract
- Line: 32: cardiovascular efficiency may not be appropriate to use for slope of the HR vs distance correlation.
- Line 34: recommend using female, not just F.
- Line 34: based on participant’s age, it is not really representing adolescents and young adults.
- Line 35: unclear what the performance is. When reporting P value like this, it is generally shown as p<0.001 not P<0.0001
- Line 37: again not sure what the number is coming from (1348.3±71.8 m)
Introduction
- Unclearly defined disability.
- Precision exercise based training would require more rigorous phenotyping or pheno-grouping of patients, risk stratification, genomic profiling based on the biologic mechanisms, as well as identifying effective exercise dose including frequency, intensity, type, time, volume and progression so more optimized exercise training can be delivered to cancer patients. Previous articles are available, which may help improve the current manuscript. Currently it may be hard to think this manuscript is addressing precision exercise interventions.
- doi: 10.1200/JCO.2015.62.7687 J Clin Oncol. 2015 Dec 10; 33(35): 4134–4137.
- https://doi.org/10.1161/CIRCULATIONAHA.117.024671Circulation. 2018;137:1176–1191
- Currently, the gold standard of maximum aerobic capacity is assessed by cardiopulmonary exercise testing (CPET). However, authors did not describe any of this concept, especially lack of justification why YoYo AD is needed rather than using the current gold standard method which is CPET. Further, it is unclear if the Yo Yo AD is valid, compared to the gold standard method.
Methods
- Line 136: it is unclear how the workload was determined by a sports medicine doctor and pediatrician. Currently, it appears that it was subjectively determined. Authors stated that there are different protocols like frail training protocol, most frail training protocol. More details may help understand how risk was stratified and assigned participants into different groups.
- Figure 1. Font size for the table is too small
- Figure 1. Title is missing
- Line 157: typo needs to be corrected
- Line 173: If 43% of participants didn’t reach the distance in the established time, is it really a feasible test?
- Line 189: recommend reporting study design upfront early in the methods section and only statistical analysis reported here. While the main focus is YoYo AD test, the sample size calculation is based on 6MWT or TUDS? Further this is 2 group x 2 time points design so it should be 2x2 Repeated measures ANCOVA for the statistical analysis, not one way ANOVA. Would recommend consultation with biostatistician.
Results
- Line 220: unclear if this is intrarater reliability of interrater reliability
- Line 240: this may not show “validity” without comparing this to the gold standard test.
- Line 256: Figur 4a. this is heart rate response, not cardiovascular response. Also Figure 4a and are very hard to see what the main effects. Would recommend removing or replacing with better visualization. For Figure 4b, how were the number assessed by which methods? (750.7±23.8 vs 1200.7±23.8 268 m, p=0.0618 (1300.3±231.9 vs 1200.7±23.8 m, p=0.9618)
Discussion
- 1. again interrater and intrarater reliability would need to be addressed.
- 2. 6MWT and TUDS would more likely represent “physical function” rather than representing phosphocreatine hydrolysis anerobic glycolysis, mitochondrial respiration and others.
- Line 355: fragility or frailty? Not consistently used in this manuscript.
Reviewer 3 Report
Dear authors, I started reading your manuscript with great interest, because I think that this field of pediatric exercise oncology has the potential to really ad some valuable research and evidence to the current state of knowledge regarding exercise and cancer.
Although, I have to admit that I was quite confused regarding the structure, abbreviations and some missing aspects in your manuscript and I try to explain that in the following.
Simple Summary and Abstract: appropriate (although I think the many abbreviations, especially the abbreviation for children, are really confusing and make the texts difficult to read)
Key words: appropriate.
Abbreviations: confusing sometimes. I understand that you try to shorten the long expressions, but I would recommend not to abbreviate children, that's uncommon, I think.
Structure: The structure of the whole manuscript is not clear to me. For me, there are some mixtures of discussion aspects in the introduction (e.g., page 2 lines 73ff as well as line 93ff. This should not be part of the introduction. in 2. Materials and Methods, the intervention is described within the paragraph 2.1 participants. I don't think, the intervention should be part of this paragraph, but should be named in an own paragraph. Furthermore, the different protocols, e.g., the standard training protocol or the most frail training protocol should be explained further. On page 4 line 170ff, is, for me, part of the results section because reasons are named to end the test. That should not be part of the Materials and Methods section. The conclusion is way to short and doesn't offer any guideline for the reader. Which patient group is able to perform this test? What do you conclude from the results for clinical exercise programs? That sentence sounds to general for me.
Introduction:
- References are missing, e.g. for the aspects on page 2 line 50, line 55, line 67, line 84 (this is for the discussion part, from my point of view). The whole paragraph from page 2 line 93 until page 3 line 102 only refers to one reference, that seems a bit weird. The following paragraph is not supported by any reference at all and in the last paragraph of the introduction, Bangsbo et al. is named without a year.
- The PEx are completely unknown to me although I work in this field for many years. Please add some more explanation on that.
Materials and Methods:
- I already named the mix up with the results section within the paragraphs.
- The figure 1 is too small and one can hardly read the numbers in the little table.
- Page 4 line 170ff again, this should be part of the results section.
- 2.3 names the assessments, but the time points for testing are not named at all. The authors talk about a baseline assessment, but I can't find a number of days post-diagnosis, for example. Please add this, I think, this is important information to the reader.
- I hope, I didn't oversee this, but did you explain Yo-Yo somewhere?
Results:
- Table 1 is missing the correct explanations, e.g., is there median in age, the standard deviation; time since diagnosis; gender; relapses etc. - I think, this huge number of participants hold the potential to describe the collective more detailed. Please add a clear and explaining legend to all tables and figures.
- For me, the other tables and results regarding statistical aspects is sufficient, but I'm not a statistician and I don't feel qualified to judge this statistical methods.
Discussion:
- Overall, please add more references to integrate your findings in the current state of knowledge. Page 10 line 301ff or page 11 line 331ff and line 339 are not including any references. This is not a discussion, but the authors opinion, maybe.
- You talk about safety and the test to be safe, but how do you evaluate safety? There's - in the methods section - named that no adverse events have been recorded. Please discuss this in the discussion section.
From my opinion, this data can add interesting knowledge to the current state of the art, but the manuscript needs major revision and a clear structure.
I hope, my comments are understandable and helpful for you.
All the best for your future research.
Reviewer 4 Report
Overall, the manuscript is written quite nicely. I only have minor suggestions:
Authors should use past tense throughout since the study has already been completed.
Authors only need to report p values with 3 decimals.
Ine 54: change “complain” to “sustain”
Line 64: change “temporary” to “transient”
Line 100: Use formal language throughout. Change “roller coaster” to “transitory”
Line 156: Was adequate hearing a question asked to participants since you are using an audio cue?
Line 159: typo
Figure 4A/B: The lime green color is extremely hard to see. Perhaps make it darker to make it easier to see on the screen.
Authors need to discussion limitations and future research at the end of the discussion.